# Casual Insights into Parler's Content Moderation Shift: Effects on Toxicity and Factuality

## Abstract

Social media platforms employ various content moderation techniques to remove harmful, offensive, and hate speech content. The moderation level varies across platforms; even over time, it can evolve in a platform. For example, Parler, a fringe social media platform popular among conservative users, was known to have the least restrictive moderation policies, claiming to have open discussion spaces for their users. However, after linking the 2021 US Capitol Riots and the activity of some groups on Parler, such as QAnon and Proud Boys, on January 12, 2021, Parler was removed from the Apple and Google App Store and suspended from Amazon Cloud hosting service. Parler would have to modify their moderation policies to return to these online stores. After a month of downtime, Parler was back online with a new set of user guidelines, which reflected stricter content moderation.

In this paper, we studied the moderation changes performed by Parler and their effect on the toxicity of its content. We collected a large longitudinal Parler dataset with 17M parleys from 432K active users from February 2021 to January 2022, after its return to the Internet and App Store. To the best of our knowledge, this is the first study investigating the changes in content moderation policies of Parler using data-driven approaches and also the first Parler dataset after its brief hiatus. Our quasi-experimental analysis indicates that after the change in Parler's moderation, all forms of toxicity saw a significant decrease ($p < 0.001$). Finally, we found an increase in the factuality of the news sites being shared, as well as a decrease in the number of conspiracy/pseudoscience sources.

## CCS Concepts

• **Information systems** → **Web mining**; **Social networking sites**.

## Keywords

Parler, Content Moderation, Hate Speech, Moderation Changes, Social Media

### ACM Reference Format:

Anonymous Author(s). 2025. Casual Insights into Parler's Content Moderation Shift: Effects on Toxicity and Factuality. In *Proceedings of The Web Conference(WWW) '25 (The Web Conference (WWW) '25)*. ACM, New York, NY, USA, 10 pages. https://doi.org/TBA

## 1 Introduction

Social media has become a powerful tool that reflects the best and worst aspects of human communication. On one hand, they allow individuals to freely express opinions, engage in interpersonal communication, and learn about new trends and stories. On the other hand, they have also become fertile grounds for several forms of abuse, harassment, and the dissemination of misinformation [11, 52, 59, 81]. Social media platforms, hence, continue to adopt and evolve their content moderation techniques and policies to address these issues while trying to respect freedom of speech and promote a healthier online environment.

Social media platforms, however, do not follow unified methods and policies for content moderation [82]. While some social media platforms adopt more stringent content moderation rules, others, like Parler, pursue a laissez-faire approach. Parler, launched in 2018, adhered to this hands-off moderation philosophy, contending that it promoted richer discussions and protected users' freedom of speech [76]. This was until January 6th, 2021, when Parler gained much notoriety for being home to several groups and protesters who stormed Capitol Hill [38, 75]. Subsequently, due to its content moderation policies and concerns about the spread of harmful or extremist content, Parler faced significant consequences. It was not only terminated by its cloud service provider, Amazon Web Services but also removed from major app distribution platforms, including the App Store and the Google Play store [35].

For Paler to return to the Apple App Store, it had to enact substantial revisions to its *hate speech policies*.[1] This included a complete removal of the ability for users on iOS devices to access objectionable and Not Safe for Work (NSFW) content. As a result, Parler's updated policies introduced more stringent moderation policies aimed at curbing hate speech on the platform [53]. While several other prior studies have focused on the impact of de-platforming users or certain communities [8, 20, 51, 70, 72, 74], or investigated how content moderation has an impact on activities of problematic users [8, 88], our work is the first that investigates the impact of stricter content moderation policies on the platform's content and its existing userbase. In particular, we investigated two research questions: **RQ1:** Did changes to Parler's content moderation guidelines had any significant impact on the user-generated content? **RQ2:** How have Parler's content moderation revisions changed its existing users' characteristics?

To assess these effects, we conducted a quasi-experimental Difference-in-Difference (DiD) analysis [5], monitoring user posts for toxic content, insults, identity attacks, profanity, and threats. In addition, we explored shifts in conversation topics and quantified the presence of biased posts and posts with non-factual links, utilizing data sourced from Media Bias Fact Check (MBFC) [2].

To answer the above research questions, we used the data from Aliapoulios et al. [9] as the seed dataset (we call this dataset a

---

[1]Example of changes in Parler CG: https://tinyurl.com/yda6pfmj

*pre-policy change* dataset), and we tried to collect the posts for the same sample of users (i.e., 4M). Using our custom build crawler, we could collect about 17M parleys of 432K active users from February 2021 to January 2022. We labeled our dataset as a *post-policy policy change* dataset. To the best of our knowledge, ours is the first dataset that was collected after Parler came back online. We will make our dataset available to the public. To measure the effect of Parler's content moderation changes, we used the Difference-in-Difference (DiD) regression analysis, which is arguably one of the strongest and widely used quasi-experimental methods in causal inference [26, 34, 42, 47]. This analysis helped us understand how and if the outcomes, e.g., the number of toxic posts, have changed after Parler changed its moderation guidelines.

Thus, in this paper, we have the following contributions and findings:

(1) Our work shows how content moderation effectiveness can be tested employing data-driven analysis on data obtained from the platform (here Parler).

(2) We collected Parler data after its return to the Internet and App Store, hence, the first-ever post-de-platforming dataset.

(3) Using DiD approach, we found that Parler was effective in removing the abusive and toxic content of users'. We observe that all the Perspective attributes had a significant decrease ($p < 0.001$).

(4) Our findings showed an increase in both follower and following counts, as well as an uptick in users with verified and gold badges. This suggests the potential growth of Parler's user base and the continued presence of older users who were active before the moderation policy changes.

(5) We observed an improvement in factuality and credibility scores from the pre-moderation dataset to the post-moderation dataset. Additionally, we noted a reduction in the sharing of conspiracy and pseudoscience source links. However, there was an increase in the sharing of questionable source links in the post-moderation change dataset.

## 2 Related Works

**Fringe Communities:** Over the past few years, scholarships have extensively studied various fringe platforms such as Gab, 4chan [16, 46, 50, 83, 93]. When compared to other fringe social media, Parler is younger. Due to this, we notice that not a lot of studies have focused on collecting or establishing a framework to collect data from Parler [10, 69]. There have been studies comparing topics of discussion on Parler and Twitter [69, 83]. Although there exists work in this domain, most of the work is focused on exploring the existence or prevalence of a single topic. Hitkul et al. [69] uses the capitol riots, a pivotal movement in Parler's history, to compare topics of discussion between Parler and Twitter. Works have analysed the language in Parler in several aspects such as QAnon content [13, 83], and COVID-19 vaccines [12]. We believe that our work differs in this aspect as we are studying the changes localized to Parler and how users reacted to a brief hiatus of Parler.

**Studies about Deplatforming:** Jhaver et al. [51] examined how deplatforming users on Twitter could impact their userbase. They found that banning significantly reduced the number of conversations about all three individuals on Twitter and the toxicity levels of supporters declined. Trujillo & Cresci [88] found that

interventions had strong positive effects on reducing the activity of problematic users both inside and outside of r/The_Donald. Some scholarships have examined the effects of deplatforming individuals on the sites that sanctioned influencers move to post-deplatforming [8, 47, 70, 72, 74, 77]. These researchers found a common result, that deplatforming significantly decreased the reach of the deplatformed users, however, the hateful and toxic rhetoric increased. Unlike previous studies, instead of exploring how users' discourse changes when switching to different platforms, we examine the same users' discourse when the platform undergoes stricter content moderation policies.

**Hate Speech Detection and Classification:** Empirical work on toxicity has employed machine learning based detection algorithms to identify and classify offensive language, hate speech, and cyberbully [28, 68]. Features including lexical properties, such as n-gram features [61], character n-gram features [58], character n-gram, demographic and geographic features [90], sentiment scores [29, 84] average word and paragraph embeddings [31, 61], and linguistic, psychological, and effective features inferred using an open vocabulary approach [32] have been used to detect hate speech. Google's Perspective API [37] has been extensively used in the previous studies [7, 32, 39, 45, 64, 78, 80, 94].

**Media Bias Fact Check:** Gruppi et al. [41] used MBFC service to label websites and the tweets pertaining to COVID-19 and 2020 Presidential elections embedded inside these articles. Weld et al. [91] analyzed more than 550 million links spanning 4 years on Reddit using MBFC. MBFC is widely used for labeling credibility and factuality of news sources for downstream analysis [18, 21, 22, 27, 33, 44, 55, 60, 85, 91] and as ground truth for prediction tasks [19, 30, 40, 67, 86].

## 3 Background & Data Collection

We will now describe in detail about Parler and also our data collection methodology.

**Parler:** Parler is a microblogging website that was launched in August 2018. Parler marketed itself as being "built upon a foundation of respect for privacy and personal data, free speech, free markets, and ethical, transparent corporate policy" [43]. Parler is known for its minimal restrictions and many right-wing individuals, citing censorship from mainstream platforms joined Parler [82]. However, after linking the 2021 US Capitol Riots and the activity of some groups on Parler, such as QAnon and Proud Boys, on January 12, 2021, Parler was removed from the Apple App Store, the Google Play Store, and Amazon Web Services [35]. Parler had to change considerably its *hate speech* policy to return to the Apple App Store [53]. Parler came back online with a new cloud service provider in early February 2021 [73]. On April 14th, 2023, Parler was bought a digital media conglomerate Starboard, and currently, is back online as rebranded *Parler 3.0* [66].

**Pre Policy Change Dataset:** The data collection tool used in Aliapoulios et al. [9] managed to collect user information for almost all of the users present at the time based on estimates published by Parler. They also managed to collect posts and comments from these users dating back to 2018 when Parler was created. The study collected user information from more than 13.25M users' and randomly selected 4M users out of these, and collected about 99M posts (or parleys), and about 85M comments from August 1st, 2018

to January 11, 2021 for these 4M users'. In our study, we call this dataset *the pre-policy change dataset*. We used these 4M users as the seed dataset to collect data post-policy changes instituted by Parler.

**Post Policy Change Parler Dataset:** Parler started to employ a different mechanism for authenticating API requests in February of 2021. Due to these changes, we could no longer collect data from all users. Hence, we first obtained the list of 4M users provided in the pre-policy change dataset for which the authors collected posts and comments [10] and used our custom-build framework to get the content posted by the same users. Using this framework, we collected information about the post body, any URLs posted, a URL to the location of any media posted, the date posted, the number of echoes, and other metadata, such as the badges of the poster. Authors in [9] obtained the metadata of 13.25M users, hence we also tried to collect the metadata of these users. We will make both our dataset and framework public.

**Post Policy Change Dataset Statistics:** From the 4M users, we could collect 17,389,610 parleys from 432,654 active users. Our dataset consists of parleys from February 1st, 2021 to January 15th, 2022. We used the */pages/feed* endpoint, which returns the parleys (posts) posted by a specific user using their username. Note that, this endpoint is different from the endpoint that is used to collect the 13.25M users' metadata, and hence we were only able to obtain 432K users' parleys. Several users from the initial seed dataset of 4M were no longer active. Manually checking these accounts we found that they had either deleted their accounts, or changed their usernames, or did not post any parley after Parler's return, or switched to private accounts. Note, we did not include any users' post if their account was private. We label them as *missing* users. Even though we are unsure if these users were suspended by Parler or they decided to leave Parler, we nevertheless analyzed and compared these users with those that remained active. Since we only collected posts from 432,654 users, we acknowledge that certain trends and analyses conducted might not be accurately reflected on the platform. However, as of January 2022, months after returning to the Apple app store, Parler disclosed that they estimate to have around 700,000 to 1M active users [3]. This ensures that we have collected a significant part of the data to study and base our findings on.

These parleys (17M) consisted of users posting around 9M links and plain text in the body. A majority of these posts were primary posts that had no parent. If a parley is an original parley and is not an echo of another parley, it is known as a primary post with no parent. We collected the full-text body, a URL if a link was shared, the title of the parley, the date of creation, flags for trolling, sensitive and self-reported, an upvoted flag, a counter of echoes and likes. We noticed that Parler has a trolling flag, which might be set manually by moderators or automatically by the platform.

We also tried to collect profile information for 13.25M users from the pre-policy change dataset. We used the *pages/profile/view* endpoint, that returns the metadata of the user. We found that 12,497,131 of these users still had a valid Parler account, so we could collect metadata for these users. For a vast majority of the accounts, Parler returned the number of followers, the number of following, status (account available or deleted), the number of and types of badges given to the user, a description of all badges

available on Parler at the time of collection, date of parley creation, whether the account is private or public, and also whether the account is being followed by or a follower of the user logged in. A minority of profiles have one or more of these fields missing due to changes on the Parler platform from when the user created the account and the time of data collection.

**Twitter Data Collection (Baseline):** Major and contentious events, such as the U.S. presidential election, can influence the toxicity in any social media platform. Since our statistical analysis cannot account for such confounding factors, we hypothesize that similar trends may be observable on other social media platforms like Twitter, which did not implement policy changes during the same period. If we observe different trends or varying effect levels, we can be more confident that the shifts in outcomes are attributable to Parler's content moderation adjustments. For a comparative baseline, we gathered a sample of Twitter data from the same timeframe and analyzed the trends before and after Parler's policy changes. To collect the data, we used Twitter V2 API [89] and collected posts daily using the exact timeline of the datasets. To circumvent the API restrictions, we collected 27K posts daily and restricted the posts to English. After the data collection was done, we were able to collect 24.16M posts for a pre-policy timeline and 9.69M for a post-policy timeline.

**Ethical consideration.** We only gathered posts from Parler profiles set to public and did not attempt to access private accounts. We used the same backend APIs that a user browser would request data from. We only obtained the random sample from Twitter API and did not collect any metadata information about the users whose profiles were set as private.

## 4 Methodology

We now describe how we operationalized users' behavior and characteristics to answer our two research questions.

**Measuring Toxicity Scores.** To understand the differences in the user discourse in the pre-moderation dataset vs. in the post-moderation dataset, we utilized the Google Perspective API, which is a state-of-the-art toxicity detection tool [37]. This AI-based tool investigates the provided text and assigns a score between 0 to 1, with a higher score indicating more severity for a particular attribute. We obtained the following attributes: *Severe toxicity*, *Profanity*, *Identity Attacks*, *Threats*, *Insults*, *Toxicity*

For our study, we collected the likelihood scores for each attribute. While collecting the scores, English was used as the default language for all posts since previous studies showed us that a large majority of Parler's userbase was using English as their language of choice to communicate with other Parler users [9]. Before sending the posts to Perspective API, we pre-processed the posts by removing URLs, hashtags, etc. as these can lead to wrong scores or errors computing the scores by the API. We also pre-processed our Twitter dataset (Baseline) and obtained the scores via Perspective API. While we acknowledge that Perspective API has weaknesses in detecting toxicity [49], however, prior works have found that it is successful in detecting forms of toxic language in the generated text [63], and also multiple prior works have used Perspective API for toxicity detection [7, 32, 45, 80], hence it is reasonable to use the API for our study.

**Assessing Bias and Factuality:** We analyzed the links that were shared on Parler using the Media Bias Fact Check (MBFC) service [2]. MBFC is an independent organization that uses volunteer and paid contributors to rate and store information about news websites [2]. MBFC can be used to measure the factuality of the URL, the presence of any bias, the country of origin, and the presence of conspiracy or pseudoscience, questionable sources, and pro-science sources. We used a list of links shared from both of our datasets to obtain labels for:

- *Factuality*: Referred to as how factual a website is. Scored between 0-5, where a score of 0 means that a website is not factual and a five is very factual. MBFC defines that for a website to be very factual and get a score of 5, it should pass its fact-checking test as well as make sure that critical information is not omitted.

- *Bias*: MBFC assigns a bias rating of Extreme left, left, left-center, least biased, right-center, right, and extreme right. To assign a bias rating to a website, MBFC contributors check the website's stance on American issues, which divides left-biased websites from right-biased websites [1].

- *Presence of conspiracy-pseudoscience*: Websites that publish unverified information related to known conspiracies such as the New World Order, Illuminati, False flags, aliens, anti-vaccine, etc.

- *Usage of questionable sources*: MBFC defines this as *a questionable source exhibits any of the following: extreme bias, overt propaganda, poor or no sourcing to credible information, a complete lack of transparency, and/or is fake news. Fake News is the deliberate attempt to publish hoaxes and/or disinformation for profit or influence.*

**Casual Inference:** We employed a quasi-experimental approach to measure the effect of Parler's content moderation change. We leveraged a causal inference strategy called Difference-in-difference (DiD) model [5]. In the DiD analysis, we track the casual effect of our dependent variable, i.e., *toxicity attributes* over time and using regression, by comparing the set of units where the event happened *treatment group* (i.e., *Changes in Parler's content moderation policies after its return*) in relation to units where the event did not happen *control group*. In our case, our treatment group is *Parler* and control group is *Twitter*. It is important to regress the behavior on time; otherwise, we could easily misinterpret a steadily increasing (or decreasing) time series as a treatment effect when comparing the averages of behavior prior to and post treatment [15]. Hence, DiD analysis allows us to claim that the toxicity changed at the time of, instead of simply reflecting a general trend. Herein we employ a linear regression model to estimate the effect of $\delta$ of the changes in Parler's content moderation policies after its return:

$$Y = \beta_1 T + \beta_2 P + \delta TP + \epsilon \tag{1}$$

where $Y$ is the toxicity score; $T$ is a dummy variable indicating the treatment (=1) and control (=0) group; $P$ is a dummy variable indicating observation collected before (=0) or after (=1) treatment. We estimate the coefficient $\delta$ associated with the interaction between the dummy variables $T$ and $P$ using OLS to obtain the average treatment effect. $\beta_1$ indicates the difference between treatment and the control group before changes in Parler's content moderation guidelines (treatment), $\beta_2$ indicates the change in the outcome over time for the control group i.e., Twitter (post treatment), and $\delta$ indicates the impact of the changes of content moderation guidelines on Parler (DiD).

We choose DiD over ITS (Interrupted Time Series) analysis because, DiD method is widely recognized in the econometrics and casual inference community for handling quasi-experimental interventions [14, 92]. Additionally, DiD yields a single causal estimand (i.e., $\delta$), which would simplify the interpretation of results, wherein ITS provides six separate estimates (three for Parler and three for Twitter).

## 5 Results

### 5.1 Impact of Changes to Parler Content Moderation

To have a balanced dataset, and since our post-moderation dataset spans approximately 11 months, we filtered the pre-moderation dataset for 11 months (i.e., February 2020 to January 2021). We also employed the same step for our Twitter dataset. After filtering the data, we clustered the data points based on the day the parley or the tweet was posted. After clustering the data per day, we set the *Perspective Score* for all the tweets' or parleys' as 0 if they were below 0.5, values above or equal to 0.5 and we kept the absolute value. We choose a threshold of 0.5, because prior research has used this threshold to distinguish if a post is toxic or not [7, 80]. We then averaged the scores per day to get a final score that we passed to our DiD regression model. To check the robustness of our method, we ran a regression model, where we did not use a threshold, and obtained the same results as we obtained when using the threshold, hence our model is robust.

Table 1 shows the results of our DiD analysis. We compared the trends that we observed in Parler to that of Twitter. Since we cannot control for various co-founding factors, we used Twitter as a baseline to understand if the changes were just local to Parler or was this a trend in other social media platforms also).

**Table 1: Difference in Difference (DiD) regression results for toxicity attributes.**

| Event | Dependent variable: Toxicity | Confidence Intervals |
|---|---|---|
| Treatment | 0.1058 (0.000)*** | [0.099, 0.113] |
| Post Treatment | -0.0028 (0.411) | [-0.010, 0.004] |
| DiD ($\delta$) | -0.0814 (0.000)*** | [-0.091, -0.072] |
| Event | Dependent variable: Severe Toxicity | Confidence Intervals |
| Treatment | 0.1173 (0.000)*** | [0.114, 0.120] |
| Post Treatment | -0.0024 (0.099) | [-0.005, 0.000] |
| DiD ($\delta$) | -0.0570 (0.000)*** | [-0.061, -0.053] |
| Event | Dependent variable: Profanity | Confidence Intervals |
| Treatment | 0.0691 (0.000)*** | [0.063, 0.075] |
| Post Treatment | 0.0012 (0.702) | [-0.005, 0.007] |
| DiD ($\delta$) | -0.0693 (0.000)*** | [-0.078, -0.061] |
| Event | Dependent variable: Threat | Confidence Intervals |
| Treatment | 0.1605 (0.000)*** | [0.157, 0.164] |
| Post Treatment | -0.0044 (0.025)* | [-0.008, -0.001] |
| DiD ($\delta$) | -0.0414 (0.000)*** | [-0.047, -0.036] |
| Event | Dependent variable: Insult | Confidence Intervals |
| Treatment | 0.1479 (0.000)*** | [0.143, 0.153] |
| Post Treatment | 0.0082 (0.000)*** | [0.004, 0.013] |
| DiD ($\delta$) | -0.0820 (0.000)*** | [-0.089, -0.075] |
| Event | Dependent variable: Identity Attack | Confidence Intervals |
| Treatment | 0.1282 (0.000)*** | [0.125, 0.131] |
| Post Treatment | 0.0087 (0.000)*** | [0.006, 0.012] |
| DiD ($\delta$) | -0.0382 (0.000)*** | [-0.042, -0.034] |
| *Note:* | *p<0.05; **p<0.01; ***p<0.001 | |

As we can observe from Table 1, after Parler came back online and instituted the changes to its content moderation guidelines, *Toxicity*, *Severe Toxicity*, *Profanity*, *Threat*, *Insult*, and *Identity Attack* all decreased significantly ($p < 0.001$). Additionally, we can observe that our *Treatment* variable, which indicates whether Parler and Twitter were different in effectively moderating our different dependent variables before the changes in guidelines, shows that Parler on average had a higher significant level of *Toxicity*, *Severe Toxicity*, *Profanity*, *Threat*, *Insult*, and *Identity Attack* than Twitter ($p < 0.001$). Interestingly, our *Post Treatment* variable, Twitter users' *Threat* posts saw a statistically significant decrease ($p < 0.05$), which can signify that after Parler came back online, there was a general trend that saw a decrease in users' threatening posts, which might negate the findings for *Threat* for Parler ($\delta$), however, if we observe for *Insult* and *Identity Attack*, we saw that there was a statistically significant increase in *Twitter* ($p < 0.001$), meanwhile we saw an opposite for *Parler* ($\delta$), hence we can conclude that while there was a general decrease in *Threat* posts, overall the changes to content moderation guidelines in Parler had a positive effect, i.e., lowering the abusive content of users. Additionally, Table 1 also shows the confidence intervals for all our variables. As we can see for all our *dependent variables*, 95% of the time we would believe that *did* ($\delta$), the effect of this on our various dependent variables, is between the various lower and upper bounds. This also clearly shows that the various differences in propositions are significant especially when investigating Parler pre- and post- moderation changes. Future scholarships can use our dataset, to understand why there was an overall decrease in *Threat* posts.

To further disentangle these findings, we plotted the results from our DiD model. Figure 1 shows the result. The red solid lines shows the $\delta$ i.e., DiD, green line denotes when Parler changed its guidelines and came back after its hiatus. As we can observe, across all the attributes, the model shows a statistically significant decrease, i.e., the red line ($\delta$) drops after the intervention is instituted in Parler. Interestingly, we can visually observe that *Profanity* (Fig. 1c), *Severe Toxicity* (Fig. 1d), and *Toxicity* (Fig. 1f) shows the biggest decrease compared to other attributes. Interestingly, we can observe that Twitter pre-intervention time series is much more variable than that of post-intervention. We found that the reason we had this variability, is due to the data that we obtained in the pre-policy timeline (24.16M) compared to 9.69M in the post-policy timeline dataset. Additionally, we also found far lesser number of posts that were toxic, i.e., that were above the 0.5 threshold. Additionally, we can observe that there is a high peak for all the perspective attributes, except for *Profanity* around day 600–620. We found that this was due to Jack Smith, being named as the special counsel in the Former President Donald Trump investigations [23, 36].

**Summary:** In summary, our Difference-in-difference (DiD) model revealed that there was a statistically significant decrease across all the Perspective attributes of Parler users' after Parler returned back online with stricter moderation guidelines (as shown in Table 1). We can observe that our *Treatment* (i.e., $\beta_1$) was positive for all the Perspective attributes, signifying that users' on Parler were on average posting more abusive content than the users' on Twitter. Additionally, we can observe that only *Threat* was the only variable that saw a statistically decrease not only in Parler but also Twitter. Hence, we can conclude that Parler's changes to the

moderation policy had a positive effect in decreasing the toxic and abusive content of its users, hence answering our RQ1.

## 5.2 Analysis on Parleys

To understand if Parler moderation change had an impact on the user base, other than users' speech, we also performed various analyses, on no. of following, followers, badges changed, the topic of conversation, and what were the changes in the *bias* and *credibility* in the URLs being shared changed or not. This analysis helped us in answering our RQ2.

**Comparing Users' Characteristics Pre- and Post-Policy Change Datasets:**

**Table 2: Comparison of Users Characteristics**

| Metric | Pre Policy Change | | | | Post Policy Change | | | |
|---|---|---|---|---|---|---|---|---|
| | Min | Max | Mean | Median | Min | Max | Mean | Median |
| Followers | 0 | 2,300,000 | 20.65 | 1 | 0 | 6,048,750 | 34.8 | 1 |
| Following | 0 | 126,000 | 28.28 | 6 | 0 | 479,412 | 33.4 | 8 |

We extracted *following* and *followers* counts from both datasets to understand if any of these metrics have changed significantly after the moderation policy changes. Since these variables are not captured in time, and we have two different distributions, we cannot perform DiD regression analysis. The resulting values did not form a normal distribution, so we used the Mann-Whitney test. We could reject the null hypothesis that users in the pre and post-policy change datasets have the same distribution for *followers* and *followings*. We found that there is an increase in the number of followings ($Med_{pre} = 6$ vs. $Med_{post} = 8$, $p < 0.0001$), hence indicating that users are still active on Parler. Interestingly, we can also observe that both *following* and *followers* increased in the post-moderation dataset. We hypothesize that these are new users' who joined Parler.

**Table 3: Most Popular Websites Shared on Parler**

| Website | Pre Policy | Post Policy | Change(%) |
|---|---|---|---|
| image-cdn.parler.com | 7,318,992 | 1 | −99.99 |
| youtube.com | 2,499,198 | 225,562 | −83.44 |
| youtu.be | 1,812,871 | 19 | −99.99 |
| bit.ly | 893,603 | 5 | −99.99 |
| twitter.com | 803,514 | 42,638 | −89.92 |
| media.giphy.com | 539,389 | 545 | −99.79 |
| i.imgur.com | 532,365 | 5,779 | −97.85 |
| facebook.com | 520,796 | 318 | −99.87 |
| thegatewaypundit.com | 469,855 | 610,512 | +13.01 |
| breitbart.com | 328,953 | 240,547 | −15.52 |
| foxnews.com | 298,285 | 136,956 | −37.06 |
| instagram.com | 168,160 | 22,932 | −75.99 |
| rumble.com | 164,949 | 744,132 | +63.71 |
| theepochtimes.com | 136,294 | 33,937 | −60.12 |
| hannity.com | 13,017 | 148,026 | +83.83 |
| justthenews.com | 50,638 | 147,984 | +49.01 |
| www.theblaze.com | 2,006 | 122,111 | +96.76 |
| www.westernjournal.com | 6,399 | 119,551 | +89.83 |
| bongino.com | 17,251 | 114,334 | +73.77 |
| www.bitchute.com | 104,462 | 87,672 | −8.73 |

We extracted badges for every user in the dataset of pre and post-moderation policy change as seen in Table 4. Interestingly, we found that the number of users with the *Private* badge has decreased

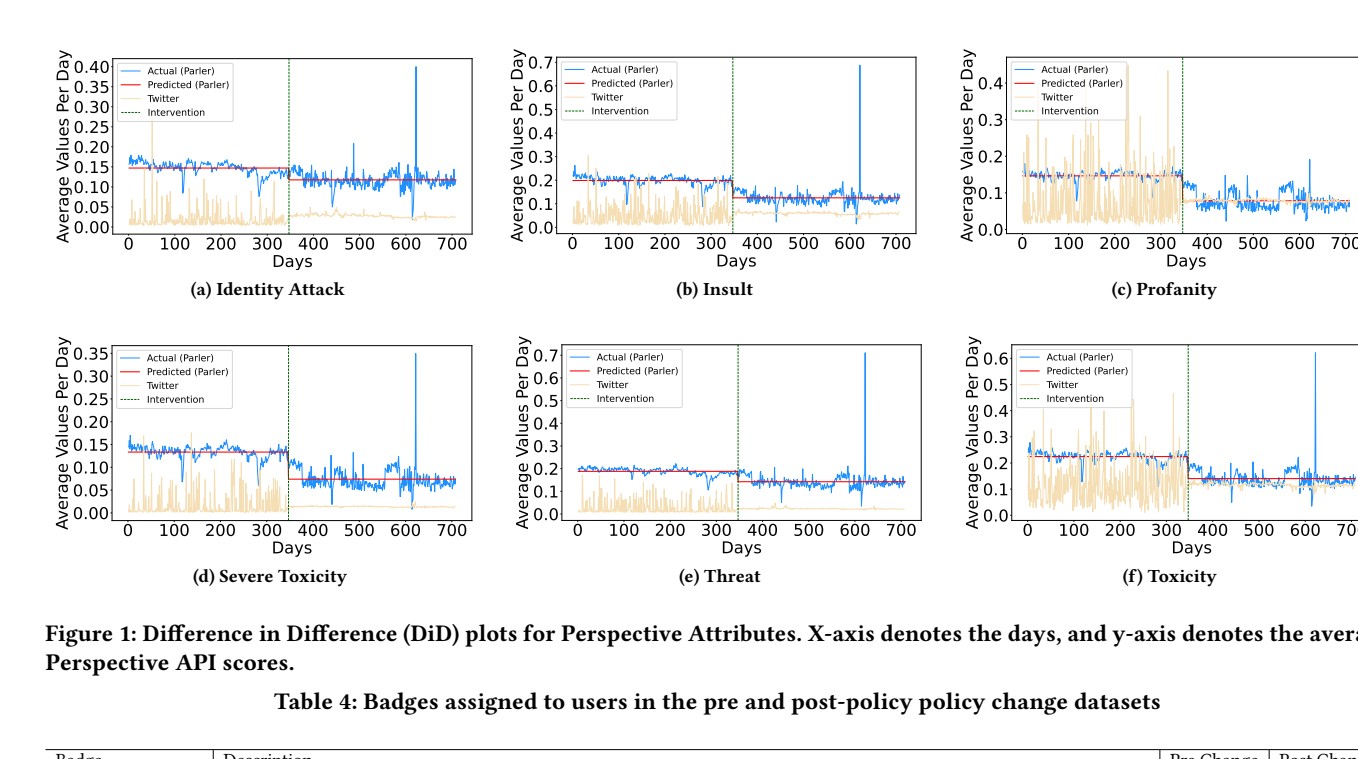

**Figure 1: Difference in Difference (DiD) plots for Perspective Attributes. X-axis denotes the days, and y-axis denotes the average Perspective API scores.**

**Table 4: Badges assigned to users in the pre and post-policy policy change datasets**

| Badge | Description | Pre Change | Post Change |
|---|---|---|---|
| Verified | This badge means Parler has verified the account belongs to a real person and not a bot. Since verified users can change their screen name, the badge does not guarantee one's identity. | 25,734 | 236,431 |
| Gold | A Gold Badge means Parler has verified the identity of the person or organization. Gold Badges can be influencers, public figures, journalists, media outlets, public officials, government entities, businesses, or organizations (including nonprofits). If the account has a Gold Badge, its parleys and comments come from real people. | 589 | 668 |
| Integration Partner | Used by publishers to import articles and other content from their websites | 64 | N/A |
| RSS feed | These accounts automatically post articles directly from an outlet's website | 99 | 13 |
| Private | If you see this badge, the account owner has chosen to make the account private. This badge may also be applied to accounts that are locked due to community guideline violations | 596,824 | 337,717 |
| Verified Comments | Users with a verified badge who are restricting comments to only other verified users. | 4,147 | N/A |
| Parody | Parler approved parody accounts. | 37 | N/A |
| Parler Employee | This badge is applied to Parler employees' personal accounts, should they wish. Their parleys are their own views and not Parler's. | 25 | 28 |
| Real Name | Users using their real name | 2 | N/A |
| Parler Early | Signifying Parler's earliest members, this badge appears on accounts opened in 2018. | 81 | 822 |
| Parler Official | These accounts - @Parler, @ParlerDev, and others - issue official statements from the Parler team. | N/A | 5 |

in the post-moderation policy dataset. Please note that we did not attempt to collect any parleys of users who had *private* badge. The badge information was extracted from the metadata of these users. We also see quite a large number of users going through Parler's verification process to prove that their account is not a bot. We notice an increase which could prove that users are still active on Parler since receiving the verified badge requires user action and is not automatic. A sharp increase in the number of verified users might be due to users' fear of an influx of bots as Parler was growing as a platform and attracting attention from other social media users. There is an increase in the number of *Gold* badges which could mean that existing Parler users have gained popularity to require a *Gold badge*.

**Parley Content Analysis:** We used textual data collected with Parleys from our data collection phase, to investigate what were users discussing about on Parler, in both the pre and post policy

change dataset. To extract the most popular topics we used the Latent Dirichlet Allocation (LDA) topic modeling technique [17]. First, we removed all the URLs any Unicode characters, and stopwords present in the text before using LDA to extract popular topics. We use a corpus of stopwords from the Natural Language Toolkit as a list of stopwords to remove from our data. We noticed a lot of interest in the 2020 US Elections in the pre-policy change dataset. This can be attributed to the fact that the elections took place during the period of pre-policy dataset collection and most news and discussions about the subject were before post-policy parleys were collected [10]. We noticed a reduction in the usage of words like *Where We Go as 1, We Go as All* (WWG1WGA). This term is associated with the QAnon conspiracy movement. We also found several Parler-specific words such as Parleys present in the earlier dataset. We theorize that the cause for this might be due to users migrating from other social media platforms like Twitter and Facebook which

do not use these terms. We also noticed Parler users were increasingly using the word *patriots*, which is how Republican lawmakers described the rioters [54].

**Links Shared in Parleys:** We examined the shared links on parleys to understand any trends, with the goal that aligning these trends with the rhetoric around online communities would allow a better understanding of the changes. To extract links being shared to external websites from Parleys, we used every Parley in both the pre-and post-policy change datasets and checked for any valid URLs being present. Then, we extracted only the top-level domain names from each URL being shared. We also stored the number of times each domain has been shared and used it to measure the popularity of websites in each dataset.

From Table 3, we can see the sharp rise in the popularity of Rumble links (64%) on Parler, as Rumble does not remove content regarding misinformation and election integrity and MBFC label this website as *Right Biased and Questionable* [57, 82]. Also, during our data collection period, the USA recently had its Presidential elections, hence we see that highlighted in our analysis. We can also notice a decline in Twitter links being shared. These observations could be explained by the sharp rise in the rhetoric surrounding censorship on Twitter and other popular social media platforms [4]. We also saw that there was a sharp increase in the number of *The Blaze* links (97%) being shared. Using the MBFC service we found that this website is labelled by the service as *Strongly Right Biased and Questionable* [56]. Observing that a large number of sites are being shared, we studied links shared on Parler using the Media Bias Fact Check (MBFC) service. We were able to collect labels for 3,937 (2.59%) and 1,081 (1.75%) of all links being shared on Parleys from the pre and post-moderation policy change datasets, respectively. We were not able to collect labels for all the URLs as the majority of them were websites such as YouTube, Twitter, and Instagram (please see Table 3) and MBFC do not provide labels for them, hence our results are generalizable, as we were able to capture a majority of the websites for which MBFC provides labels and hence, understanding how the policy change had an impact on users' speech.

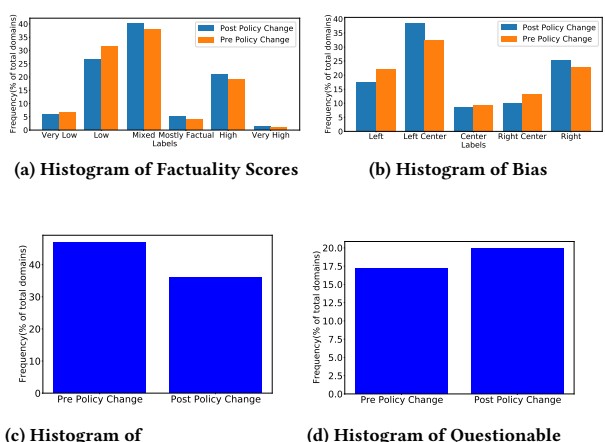

**(a) Histogram of Factuality Scores**

**(b) Histogram of Bias**

**(c) Histogram of Conspiracy-Pseudoscience**

**(d) Histogram of Questionable Sources**

Figure 2: Histogram of MBFC Labels

Figure 2 shows our results. At first, we noticed a decrease in the number of conspiracy-pseudoscience news articles in Figure 2c. However, interestingly, we saw an increase in the number of *questionable source* articles that were being shared in the post-policy change dataset, as shown in Figure 2d. This implies that Parler still does not remove URLs that are spreading overt propaganda and fake news, hence concurring with the findings of [82]. In Figure 2a we also notice that most links with a score between *Very Low* and *Low* were from the pre-policy change while post-moderation links are scattered across higher ranges between *Low* and *High*. Interestingly, in Figure 2b, we see that Parler users are sharing more URLs, from the *Left Center & Right* websites. This is interesting, as the majority of Parler users are highly conservative [24]. In summary, using the labels returned by MBFC, we found that the credibility (factuality) of the URLs being shared did increase. We also notice a substantial decrease in the number of conspiracy-pseudoscience news articles. This is interesting, as in 2022, there were notable conspiracy theories such as mpox (monkeypox), orchestrated by vaccine manufacturers, that Bill Gates was involved in the outbreak, is transmitted solely via sexual interactions, and that the WHO released the virus to gain more power [6, 95]. However, interestingly Parler users were now sharing more questionable source URLs than before.

**Summary** In summary, we observed that there was a statistically significant increase in the number of followings after Parler came back online. We also found that Parler users' *Verified* their accounts more than in the pre-moderation change dataset. Interestingly, we found that Parler users were increasingly using the word *patriots*. Using MBFC, we found that credibility (factuality) of the URLs being shared did increase. We also notice a substantial decrease in the number of conspiracy-pseudoscience news articles. However, interestingly Parler users were now sharing more questionable source URLs than before. Hence, there were considerable changes in Parler users' hence answering our RQ2.

## 6 Discussion

Our results indicate a positive impact of the changes to content moderation guidelines that Parler instituted after its ban. Our quasi-experimental analysis revealed that, after Parler instituted the changes, all the Perspective attributes saw a statistically significant decrease ($p < 0.001$). Additionally, observing from Table 1, we can observe that *Severe Toxicity*, *Threat*, and *Identity Attack* saw the biggest decrease compared to other attributes. This is interesting, as this is in direct contrast with what prior studies have found i.e., an increase in toxic rhetoric of users. Our research on the other hand highlights that when Parler changed the guidelines, the existing users' toxic rhetoric decreased. Using MBFC, we found that the *credibility* (actuality) of URLs shared by users on Parler, increased, which is in direct contrast to what was observed in [88].

**Effectiveness of policy change.** Scholarships have widely studied the effect of deplatforming on users and its effect on the user's toxic rhetoric [8, 51, 88]. Our research, however, sheds light on when a platform (i.e., Parler) has to change its content moderation policies to be allowed back online. Additionally, our research shows how the users who were active on Parler before it was taken offline and after Parler returned online, their toxic rhetoric decreased

significantly (please refer to Table 1), and how the credibility of links that were shared pre- and post policy change increased, a direct contrast to the findings in [88]. Furthermore, our results also show broadly, how changes to content moderation policies can effectively decrease the toxic rhetoric of existing users. However, some users migrated from Parler to other fringe social media such as Rumble, Gab, and Telegram, and scholarships have found that they became more active on these fringe platforms [47], which raises concerns about how platforms need to effectively design their moderation strategies in times of social unrest, as *one size cannot fit all* the situations [25, 82]. Furthermore, there can be an unintended effect where users might become more toxic on these fringe platforms [48], hence there is a need for collective and simultaneous actions to reduce the toxic rhetoric of these users as echoed by [47]. Future endeavors must investigate how to balance effective content moderation while still keeping with some platform's guarantee of First Amendment protections can be devised.

**Importance of this study.** To the best of our knowledge, ours is the first study that provides the social computational community with (a) a first-ever Parler dataset after its return, and (b) a framework that could be utilized in obtaining data from Parler. Our dataset provides a unique opportunity for computational social scientists to study users' behaviors, interactions, and topics discussed among such an understudy group of people with specific mindsets. Some important topics that can be explored from our dataset, which would be of interest to the wider community are: what type of misinformation was prevalent, understanding vaccine hesitancy of Parler users, since our dataset was collected around the same period, etc. Furthermore, Parler was bought by Starboard in 2023 and was shut down the same day [71], however, Parler is now back online and re-branded itself as *Parler 3.0* [66] with new changes to its content moderation guidelines [65] hence, our dataset as well as the framework, provides the perfect opportunity for scholarships to audit the changes of policy for Parler, from its inception in 2018 (Parler 1.0) to its first policy change in 2021 (Parler 2.0) to its current state i.e., Parler 3.0. Additionally, Singhal et al. [82] found that Parler did not have any form of *soft moderation* intervention. However, the new Parler 3.0 content moderation guidelines have introduced a form of soft moderation called *Time Out* [65]. Hence, our dataset can be used to study the effectiveness of *Time Outs*.

**Limitations & Future Work** In our current dataset, i.e., the post-policy change dataset, we could not collect a random sample of users, hence our analysis might not yield a full-scale impact of the moderation policy change. The other limitation of our work is that users might have changed their usernames when Parler was reinstated back online, to evade possible detection. We also acknowledge that Google's Perspective API as a toxicity detection also contains certain limitations and biases [62, 79, 87]. Furthermore, our work also does not capture the impact of users' hateful rhetoric when they moved to other platforms after Parler was taken offline. In the future, we plan to study the user comments on posts to understand if the moderation changes are being reflected in the comments, as comments can also shed light on the moderation changes that Parler undertook.

## 7 Conclusion

On January 12, 2021, Parler was removed from Apple and Google App Stores and Amazon Web Services, stopped hosting Parler content shortly after. This was blamed on Parler's refusal to remove posts inciting violence following the 2021 US Capitol Riots. Parler was eventually allowed back after they strengthen their moderation to remove hateful content. Our study investigated the effect of these policy changes on the user discourse by comparing users' rhetoric in pre- and post-policy change datasets.

Our quasi-experimental analysis indicates that after the change in Parler's moderation, all forms of toxicity saw a significant decrease ($p < 0.001$). Finally, we found an increase in the factuality of the news sites being shared, as well as a decrease in the number of conspiracy or pseudoscience sources being shared.

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
