# OpenReview forum: "Casual Insights into Parler's Content Moderation Shift: Effects on Toxicity and Factuality"
_ACM.org/TheWebConf/2025/Conference — WWW 2025 Poster_

### Official Review · Reviewer_qAi4 · 2024-11-12

**Novelty:** 5
**Technical Quality:** 6

**Review:**

I enjoyed this paper. I think that it is a unique finding that warrants attention from the Web research community at large. However, I am not convinced that this contribution goes beyond a poster level contribution. The majority of the paper is focused on methodology with only one paragraph focused on the importance of its contributions. This would be appropriate if the described methodology was unique. I think that this paper has the potential to be accepted with revision. Namely, more time spent on explaining why a platform being reinstated is consequentially different from previous de-platforming studies. Additionally, why is this important? Who exactly is impacted by these results? Again, this is an excellent concept that just needs to be fleshed out.

**Questions:**

Questions partially answered:
What is your definition of toxicity (beyond that’s what Google API says) ?
What are the broad impacts of toxicity?
What previous work does this build off of?
Work has been done on deplatforming. So what is so unique about users on the same platform coming back after a platform is reinstated? Why is that important?
Why should we care?

Questions unanswered:
How does this contribution go beyond just a poster level contribution?
Does the changes occur because of new content moderations or other social forces (does your methodology appropriately isolate content moderations)
How did you account for potential researcher bias?

**Reviewer Confidence:**

3: The reviewer is confident but not certain that the evaluation is correct

**Scope:**

3: The work is somewhat relevant to the Web and to the track, and is of narrow interest to a sub-community

---

### Official Review · Reviewer_eC26 · 2024-11-16

**Novelty:** 3
**Technical Quality:** 3

**Review:**

Strengths:

The paper is the first to study user behavior changes on Parler, a unique platform, following content moderation policy changes. The paper proposes a quantitative evaluation framework for content moderation effectiveness, laying a foundation for future research. Additionally, it quantifies the credibility of shared links using the Media Bias Fact Check (MBFC) service and employs statistical methods to validate hypotheses, ensuring the reliability of findings. The analysis highlights the reduction in toxic content and an increase in the credibility of shared information, addressing critical questions about the effectiveness of content governance on social media.

Weaknesses:

The related work and discussion sections lack a broader theoretical foundation. The analysis of mechanisms underlying content moderation policies and their applicability is shallow, focusing only on data-driven insights without offering concrete policy recommendations.

While the study observes a decrease in toxic content, it also notes an increase in sharing links from questionable sources, which seems to contradict the effectiveness of the policy changes. The paper does not adequately explain this inconsistency or explore the contextual significance of these links.

The paper relies on the Google Perspective API as the primary toxicity detection tool but does not sufficiently address its potential instability and biases. Specifically:

The Google Perspective API may produce inconsistent results, especially with complex or implicit toxic content.

There is no mention of whether the authors performed consistency checks through repeated testing or validated the toxicity scores through random manual sampling. Such validation is crucial for ensuring reliability, particularly when dealing with social media content that may involve contextual metaphors or cultural nuances.

**Questions:**

1.Why was Twitter chosen as the control group? Given the significant differences in user demographics and platform policies between Twitter and Parler, could this affect the accuracy of causal inferences? Did the authors consider using Gab or other fringe platforms as potentially more comparable control groups?

2.Regarding the known biases of the Google Perspective API (e.g., difficulty in detecting implicit toxicity), what measures were taken to mitigate its impact on the results? Did the authors use random manual sampling to verify the toxicity scores generated by the API? If so, how was this conducted, and what were the results?

3.In the credibility analysis, the MBFC label coverage is low (1.75%-2.59%). How can the conclusions be considered broadly applicable? Over the 11-month analysis period, were there any political or social events (e.g., elections) that could have significantly influenced user behavior?

4.Did the authors consider conducting a more granular temporal analysis to distinguish between the effects of policy changes and those of external events?

5.The paper mentions an increase in the proportion of links from questionable sources. Does this suggest that users may be employing counter-strategies to resist content moderation? Could this lead to the emergence of new forms of toxicity propagation?

**Reviewer Confidence:**

4: The reviewer is certain that the evaluation is correct and very familiar with the relevant literature

**Scope:**

4: The work is relevant to the Web and to the track, and is of broad interest to the community

---

### Official Review · Reviewer_xYy4 · 2024-11-29

**Novelty:** 3
**Technical Quality:** 2

**Review:**

This paper presents a data-driven investigation of Parler's content moderation policy changes following its de-platforming in 2021, with a quasi-experimental Difference-in-Difference (DiD) analysis, to assess changes in toxicity and the factuality of shared content. Below are detailed comments categorized into major and minor points.

Major:
1. User migration analysis: the paper shows that some users migrated to other platforms, but it does not explore the impact of this migration on toxicity trends. Could conduct a cross-platform comparison by collecting data from these alternative platforms.
2. Discussion about bias in toxicity detection: the Google Perspective API, which may have limitations and biases in detecting toxicity, especially in fringe platform contexts, need more discussion about this.

Minor:
1. Representativeness: the paper note that only 432k users were active in the post-policy change dataset, a subset of the original 4M users. Address how this reduced dataset might affect the generalizability of the findings.
2. Annotating key events (e.g., when Parler came back online) in a DiD chart.

**Questions:**

1. The post-policy change dataset represents only a subset (432k users) of the pre-policy change users (4M users). How confident are you that this subset adequately represents the overall user base, especially given the possibility of systematic differences between active and inactive users? Could you provide any additional analysis or reasoning to support the representativeness of this dataset?
2. Did you observe any significant changes in metrics such as user posting frequency, likes, or echoes in the post-policy change dataset? Could these engagement trends have implications for your findings on user activity and content quality?

**Reviewer Confidence:**

3: The reviewer is confident but not certain that the evaluation is correct

**Scope:**

3: The work is somewhat relevant to the Web and to the track, and is of narrow interest to a sub-community

---

### Official Review · Reviewer_byje · 2024-12-02

**Novelty:** 4
**Technical Quality:** 4

**Review:**

This paper examines the changes in moderation by Parler and their impact on the harmfulness of content. The study confirmed that the factual accuracy of shared news sites increased, while the number of sources disseminating conspiracy theories and pseudoscience decreased.

This research is notable as the first to use a dataset from Parler following its suspension, and it provides a valuable framework for extracting data from the platform. However, the finding that stricter policies naturally suppress the toxicity of content feels somewhat expected, making its contribution to social science insights slightly limited.

**Questions:**

What types of articles increased in proportion after the policy changes? In particular, I am curious about the content of suspicious source articles that became more prevalent compared to before.

**Reviewer Confidence:**

3: The reviewer is confident but not certain that the evaluation is correct

**Scope:**

3: The work is somewhat relevant to the Web and to the track, and is of narrow interest to a sub-community